# Observation of Genetic Gain with Instrumental Insemination of Honeybee Queens

**DOI:** 10.3390/insects14030301

**Published:** 2023-03-21

**Authors:** Ségolène Maucourt, Andrée Rousseau, Frédéric Fortin, Claude Robert, Pierre Giovenazzo

**Affiliations:** 1Department of Biology, Vachon Pavillon, Université Laval, Québec, QC G1V 0A6, Canada; 2Centre de Sciences Animales de Deschambault, 120A Chemin du Roy, Deschambault, QC G0A 1S0, Canada; 3Centre de Développement du porc du Québec, 450-2590 Boulevard Laurier, Québec, QC G1V 4M6, Canada; 4Department of Animal Science, Institut sur la Nutrition et les Aliments Fonctionnels, Université Laval, Québec, QC G1V 0A6, Canada

**Keywords:** selection program, genetic gain, selection pressure, instrumental insemination, directed fertilization, breeding value, hygienic behavior, spring development, honey production

## Abstract

**Simple Summary:**

To control the reproduction of honeybees is complex due to its reproductive particularities; indeed, the mating of the queen takes place while it is in flight with several males. This particularity of the species is an additional challenge in beekeeping when it comes to succeeding in its breeding and obtaining consequent genetic gains in selection. For many years, several methods of mating control have been developed with varying efficiencies. In this study we compared the genetic gains of several performance traits of colonies (i.e., honey production) as a function of selection pressure on reproduction: either directed fertilization or instrumental insemination. The results of this study show that the genetic gains obtained in colonies with artificially inseminated queens are similar to or lower than the genetic gains obtained in colonies with naturally fertilized queens, depending on the performance traits. Overall, these results do not demonstrate that the use of instrumental insemination is effective in increasing genetic gains; however, they show that instrumental insemination can be a useful and effective tool to achieve total reproductive control within a genetic selection program.

**Abstract:**

Controlling mating in the honeybee (*Apis mellifera*) is part of one of the greatest challenges for the beekeeping industry’s genetic selection programs due to specific characteristics of their reproduction. Several techniques for supervising honeybee mating with relative effective control have been developed over the years to allow honeybee selection. As part of this project, we compared the genetic gains for several colony performance traits, obtained using the BLUP-animal method, according to the selection pressure applied in controlled reproduction (directed fertilization versus instrumental insemination). Our results show similar genetic gains for hygienic behavior and honey production between colonies whether queens were fertilized naturally or via instrumental insemination, as well as similar or lower genetic gains for colonies with queens inseminated for spring development. In addition, we noticed greater fragility in queens following insemination. These findings show that instrumental insemination is an effective tool for reproductive control in genetic selection and for estimating breeding values more precisely. However, this technique does not result in queens of superior genetic quality for commercial purposes.

## 1. Introduction

Establishing honeybee genetic improvement programs based on scientific data will help to ensure the sustainability of the beekeeping industry. Through a rigorous selection process, honeybee colonies intended for commercial operations can be selected for increased population growth, honey production and resistance to certain pathogens or parasites [1,2,3]. Currently, the most advanced beekeeping selection programs are based on a genetic evaluation of individual colonies and their breeding values, calculated using the BLUP-animal statistical model (Best Linear Unbiased Prediction) [4]. This model involves estimating the genetic component of a colony’s performance (its breeding value) based on the performance of its parents and their degree of relatedness, while eliminating identified environmental (non-genetic) effects [5]. This statistical method can improve the selection response by up to 63% in some animal production programs [6]. Beekeeping selection programs based on this model have also demonstrated significant improvement for several zootechnical colony traits, such as honey production and aggressiveness [7,8,9,10].

This practice is relatively new to the beekeeping industry but has long been used in livestock production. Indeed, the BLUP-animal statistical model has been applied for cattle selection since the 1950s [5], but only since 2007 for bees [4]. This delay is largely due to the complex genetic and reproductive characteristics of the honeybee, which were difficult to transpose to the BLUP-animal statistical model [4,11]. Among these characteristics, the natural mating behavior of bees represents a unique challenge for selection. Queens mate in flight with many drones from various genetic sources [12]. The success of a breeding program therefore largely depends on controlling queen mating [13,14]. Several mating control techniques have been developed for queen breeding and selection: isolated fertilization, directed fertilization or instrumental insemination [15]. Isolated fertilization involves moving the queen and drone fertilization site several kilometers away from all other colonies to avoid the presence of unselected drones. The islands, often used for this method, allow complete control success when situated more than 9 km from the coast [16,17]. In directed fertilization, the mating site is saturated with males from the selected colonies to guarantee maximum fertilization of the queens with the desired males [18]. These methods remain approximate, since although they enable a satisfactory success rate of mating control, they cannot provide the same certainty as instrumental insemination [17,19,20]. The latter usually consists of injecting 8 to 12 microliters of sperm from selected drones into the vagina of a queen, thus controlling fertilization. However, the beekeeping industry has been slow to adopt this technique, in particular because of the perception that instrumentally inseminated queens show poorer zootechnical performance compared to naturally fertilized queens. As a result, it has been practiced mainly for research purposes [17,21,22]. Today, with improvements in both instrumentation and method, the technique has become more efficient and reproducible [22,23].

The Canadian beekeeping industry now faces several important issues. First, Canadian beekeeping has experienced high winter mortality since 2007, with average losses of 26% per year [24,25]. These abnormally high losses have multiple causes, including parasitism and disease, depletion of floral resources, exposure to pesticides and stress associated with pollination services [24,26,27]. To address this problem, beekeepers must import bees and queens from abroad, mainly from Australia, New Zealand, Chile and the United States (California or Hawaii) [2,28]. Imported bee strains are not adapted to northern climates or Canadian beekeeping management practices [29,30,31,32]. In addition, recent events related to COVID-19 have significantly disrupted the supply of bees and queens to Canada, which has demonstrated the risks involved in relying on imports and the urgency of finding solutions [33]. Importing honeybees jeopardizes self-sufficiency efforts, food security and the sustainability of the domestic beekeeping industry.

In this study, we tested the efficacy of instrumental insemination and compared the genetic gains obtained after one year, according to the selection pressure applied through natural mating versus instrumental insemination. The specific objective was to determine if the introduction of instrumental insemination using the BLUP-animal selection model in our breeding program would accelerate genetic gain at the F1 generation for three traits of interest to the Canadian beekeeping industry: hygienic behavior, honey production and spring development.

## 2. Materials and Methods

### 2.1. Honeybee Colonies

Our study was carried out with colonies from our university honeybee research institute, Université Laval-Centre de Recherche en Science Animal de Deschambault (UL-CRSAD), Québec, QC, Canada (N 46°40.27′, W 10°71.50′). The UL-CRSAD honeybee selection program, established in 2010, uses the BLUP-animal statistical model to calculate breeding values and a selection index to choose breeder colonies each year. All colonies in the breeding program come from a common ancestral stock made up of hybrid Italian stock from local breeders in Quebec, QC, Canada and from imported Buckfast lines from Denmark (Buckfast Denmark, https://buckfast.dk/index.php/en/, accessed on 19 March 2023). Details of our breeding and selection procedure are described in Maucourt et al., 2020 [34].

For this project, we selected UL-CRSAD colonies with the highest selection indexes, which were obtained by combining the breeding values of three selection criteria: hygienic behavior, honey production and spring development, at a percentage of 50, 30 and 20, respectively. These criteria are all important traits for Canadian beekeeping. We have decided to give weight to these criteria according to the economic importance for the beekeeping industry [2,13,34]. Two selected colonies produced 30 queen sisters each, one selected colony produced the drones for sperm collection / instrumental insemination and ten selected sister colonies produced many drones for our mating station. Young virgin queens were then either instrumentally inseminated (N = 30) or naturally mated (N = 30) and separated into two different groups (Figure 1). This experimental breeding plan was replicated twice, once in 2018 and once in 2019, and the performance of resulting colonies was subsequently evaluated for each selection criterion, in 2019 and 2020, respectively.

### 2.2. The Choice of Colonies for Breeding

In 2018, the two queen-producing mother colonies for lines 1 and 2 had a selection index of 178.5 and 130.2, respectively. In 2019, the two queen-producing mother colonies for lines 3 and 4 possessed a selection index of 154.7 and 153.4 (details on the breeding values associated with the selection index are presented in Table 1). These mother colony selection indexes, composed of three breeding values, are also presented in Table 1 and demonstrate the superior quality of these colonies for breeding. For example, for the mother colony of line 1, the estimated breeding values are 185.9 for hygienic behavior, 116.3 for honey production and 133.9 for spring development. This means that the colony is genetically superior by 85.9% for hygienic behavior, by 16.3% for honey production and by 33.9% for spring development compared to the standard of the colony population in the UL-CRSAD breeding program.

Selecting drone-producing colonies is a more complex process than selecting queen-producing colonies. Indeed, using the BLUP-animal statistical model requires all drone-producing colonies within the program to be related [4]. Therefore, colonies are not selected individually as they are for queen-producing colonies, but rather as a group of several related colonies (i.e., colonies that all have sister queens). The choice of the group is based on the selection index of the colony that produced the group of related colonies, that is, the mother colony of the queens of these colonies, who is also the grandmother of the drones that we produced for this project. The choice of the group of drone-producing colonies also depends on the size of the groups. Indeed, the UL-CRSAD selection program requires a minimum of 6 drone-producing colonies for optimal fertilization of virgin queens produced during the season [10,35]. In 2018, 6 colonies were selected for drone production and the selection index of the mother colony was 104.3. Then in 2019, 8 colonies were selected for drone production and the selection index of the mother colony was 148.2 (details on the breeding values associated with the selection index are presented in Table 2).

For instrumental insemination, drones from a single drone-producing colony in the group of related colonies were used. For each of the two years, we selected the colony with the highest selection index from the group of related colonies. The drone-producing colony used for insemination in 2018 had a selection index of 106.4, and the one used for insemination in 2019 had a selection index of 136.0 (details on the breeding values associated with these selection indices are presented in Table 3).

### 2.3. Instrumental Insemination

#### 2.3.1. Semen Collection

To produce surplus drones, queens of the selected mother colonies were isolated in a queen excluder cage containing a drone frame (home-made, adapted from the following model: Ylega, Faenza, RA, Italy, 237021) for 48 h (Figure 2). A few days before the emergence of the young drones, the drone frame was placed in a small hive (dimensions: 50.8 × 17.8 × 27.9 cm) containing a frame of honey, a frame of pollen, a frame of brood with its adherent bees and a new queen. This drone colony was placed in a 42 m^2^ greenhouse tunnel covered with a market garden shade net (Figure 2) where drones could fly freely and defecate, thus avoiding contamination of semen and reducing queen infections after insemination [22,36]. The semen was collected the day before insemination, using a Harbo syringe (Harbo Large Capacity Syringe, Model GS 1100, Fisher Scientific Ltd., Ottawa, ON, Canada) and via manual eversion of the drone’s genitalia [23,37]. A sterile modified Kiev solution (0.3 g of D+ glucose, 0.41 g of potassium chloride, 0.21 g of sodium bicarbonate and 2.43 g of sodium citrate in 100 mL of distilled water) was used to lubricate the syringe tip, and 0.05% dihydrostreptomycin was added as a bactericide. The semen of approximately fifty drones was stored in 50 microliter glass capillary tubes (Microcapillary tube Drummond Microcaps^®^, P2174) in a dark room at 22 °C until the next day [38,39,40].

#### 2.3.2. Insemination of Virgin Queen

Thirty queens (N = 15 for each selected mother each year) were instrumentally inseminated following the method described by Cobey et al., 2013 [23] and with the modifications described here. Virgin queens were inseminated 5–6 days after emergence of royal cells in closed double Langstroth four-frame nuclei (Propolis-etc. …, Saint-Pie, QC, Canada; DN-1000) containing two frames of brood with their adherent young bees, an empty frame and a frame of food with honey and pollen. Each queen received four minutes of a CO_2_ anesthesia treatment 24 h prior to insemination to stimulate the production of juvenile hormones and promote the migration of spermatozoa into the spermatheca [41,42]. Each queen received 10 microliters of mixed semen using a Harbo syringe (Harbo Large Capacity Syringe, Model GS 1100, Fisher Scientific Ltd., Ottawa, ON, Canada).

After insemination, the queen was placed in a push-in cage (Propolis-etc. …, St. Pie, QC, Canada; CC-5010) with 10 emerging bees, on a brood frame in the center of her initial nucleus colony. Sugar water (1:1) scented with anise essential oil was applied to the frame’s top bars to facilitate queen acceptance [23]. After 7 days, laying queens were released in their colony.

#### 2.3.3. Natural Mating

Thirty royal cells (N = 15 for each selected mother each year) were introduced in mating nucleus colonies (dimensions: 30.5 × 17.8 × 20.3 cm) containing 2 frames of brood with young adherent bees, an empty frame and a frame with honey and pollen. The mating nucs were situated at 1200 m from selected drone-producing mother colonies and 1600 m from a previously identified drone congregation area [34,43]. After fifteen days, the egg-laying pattern and the general appearance of the queen were examined to confirm normal and uniform laying before collecting the queen, clipping half of one of her two anterior wings and marking her on the thorax with a marking pen (Propolis-etc. …, Saint-Pie, QC, Canada; MP-1103 to MP-1104). Each queen was then placed in a Jz-Bz cage with a slow-release sugar candy plugged opening (Propolis-etc. …, Saint-Pie, QC, Canada; QC-1111) with 4 accompanying workers and introduced centrally in double nuclei Langstroth 4 frames (Propolis-etc. …, Saint-Pie, QC, Canada; NU-2002) comprising 2 full brood frames with their young adherent honeybees, an empty frame with drawn cells and a food frame with honey and pollen.

#### 2.3.4. Colony Management

In September, each nucleus colony containing the inseminated and naturally mated queens was treated against *Varroa destructor* (2019 with Thymovar^®^ and 2020 with Apivar^®^), fed 10 L of sucrose–water solution (2:1) using a double nuclei feeder with floaters (Propolis-etc. …, Saint-Pie, QC, Canada; FE-t1700) and overwintered in an environmentally controlled room (4 ± 1 °C and 40–50% RH) from mid-November until mid-April (2019 and 2020).

The following year, in early May, the queens, bees and frames of each nucleus colony were transferred into Langstroth 10-frame hives. Colonies were then randomly and equally distributed in two different apiaries. These apiaries were situated within a radius of 40 km of our research center UL-CRSAD and at least 3 km apart from each other in a similar agricultural environment with the same potential honey production [10]. These colonies were managed for honey production, and their performance was evaluated during summer (the year following their conception, in 2019 and 2020, respectively).

#### 2.3.5. Colony Performance Criteria

The performance criteria were selected based on those identified by the Canadian beekeeping industry in recent years [2]. These traits are associated with health, productivity and hardiness; they are explained in Maucourt et al., 2021 and briefly described here:Hygienic behavior: we used the freeze-kill brood test [44,45] to count the percentage of killed brood within a 5cm diameter circle that were removed by bees after 24 h.Honey production: honey supers of each colony were weighed before and after summer on a platform scale (CAS-131 USA, East-Rutherford, NY, USA; CAS CI-2001BS) to calculate the colony’s total seasonal honey production.Spring development: colony strength was measured in early June, in the evaluated area occupied by immature worker honeybees (eggs + larvae + capped brood) [46,47].

#### 2.3.6. Statistical Analysis

Statistical tests to measure breeding values and selection indices were conducted using ASReml-R software (ver. 4.1.0.143, VSNi Inc., 237 England, UK) with the database containing the performance associated with colonies in our breeding program and described in Maucourt et al., 2021 [10]. All dependent variables were tested for normality using the Skewness and Kurtosis test, and no Box–Cox power transformations were required to meet the assumptions of model normality. Breeding values of colonies were estimated with the BLUP-animal model adapted to honeybees.

An ANOVA statistical test to compare the breeding values of the three performance traits between experimental groups was performed using RStudio software (ver. 2021.09.2+382, RStudio enterprise, Boston, MA, USA). A Student’s t test for an independent sample was used to compare breeding values for the two breeding lines, which were either instrumentally inseminated queens or naturally mated queens in 2019 and 2020.

## 3. Results

The total colony losses during the two years, 2019 and 2020, were 23.4% for colonies with instrumentally inseminated queens and 18.0% for colonies with naturally fertilized queens (Figure 3). For colonies with instrumentally inseminated queens, 18.6% were related to winter losses and 5.7% to summer losses, while for colonies with naturally fertilized queens, 14.3% were associated with winter losses and 3.7% with summer losses. The identified causes of summer losses (Figure 3) of colonies with instrumentally inseminated queens compared to naturally mated queens were supersedure (37% vs. 17%), presence of chalkbrood (27% vs. 41%), queenless (18% vs. 42%) and drone-laying queen (18% vs. 0%). Colonies that showed signs of chalkbrood (i.e., presence of mummies) were immediately removed from the apiaries, as well as the entire project, to avoid contamination of other colonies and a bias in their performance.

The breeding values for the three performance traits are shown in Figure 4 for each year, comparing colonies with inseminated or naturally mated queens for the two selected lines.

No significant difference was detected for hygienic behavior between lines with inseminated versus naturally mated queens in 2019 or 2020 (line 1: *t* = 1.4521; df = 16; *p* = 0.1658; line 2: *t* = −0.8507; df = 10; *p* = 0.4149; line 3: *t* = 1.1816; df = 22; *p* = 0.25; line 4: *t* = 0.6187; df = 18; *p* = 0.5439). No significant difference was detected for honey production for line 1 colonies in 2019 (*t* = 0.1534; df = 16; *p* = 0.88) or for line 3 and 4 colonies in 2020 (*t* = 1.489; df = 19; *p* = 0.1529 and *t* = 2.0597; df = 18; *p* = 0.05419, respectively). Line 2 colonies had insufficient survival rates to permit analysis due to a high rate of loss of the inseminated queens of this line. No significant difference was detected for spring development between lines with inseminated versus naturally mated queens for lines 1 and 2 in 2019 (*t* = 0.4484; df = 18; *p*= 0.6592 and *t* = −0.038; df = 12; *p* = 0.9703, respectively) and for line 4 in 2020 (*t* = 1.7571; df = 21; *p* = 0.09348). However, for line 3 in 2020, the breeding values of colonies with naturally mated queens were significantly higher than the breeding values of colonies with inseminated queens (*t* = 2.3906; df = 26; *p* = 0.02436).

## 4. Discussion

The main objective of this project was to test the efficiency of instrumental insemination and determine if it could be successfully used within our breeding program using the BLUP-animal selection model to accelerate and rapidly increase genetic gain of three traits of interest to the Canadian beekeeping industry: hygienic behavior, honey production and spring development.

Our results do not show that instrumental insemination increases the genetic gain after the first generation. This was confirmed during two consecutive years for the three performance traits measured. Indeed, we have shown that the breeding values of colonies with inseminated versus naturally fertilized queens were similar for hygienic behavior and honey production, but also similar or significantly lower for spring development.

Our results are interesting, as they question the value of using instrumental insemination compared to a well-prepared, natural mating site. In a selection program where natural mating is used, it is often difficult or impossible to obtain paternal genealogical information about queens because of their polyandrous reproductive strategy [14,48]. To overcome this difficulty, the BLUP-animal methodology integrates a fictitious father into the pedigree, which corresponds to the genealogical information of the parent colony producing the drones [4,15]. This statistical adaptation of the BLUP genetic model for honeybee selection makes it possible to obtain a more precise estimate of breeding values compared to a model that neglects paternal contributions [49]. Instrumental insemination with drones from the same selected colony represents another option. With this approach, a “father” colony can be designated in the statistical modeling, thereby obtaining more precise breeding values but rapidly increasing inbreeding coefficients [9].

The aim of our study was to measure the impact of one year of selection with instrumental insemination. In a future study, a next step could involve repetitive inbreeding cycles with the same mother line and her daughters. Three to five inbreeding cycles could be carried out during a single summer if new generations of queens are produced immediately after the first laying of the mother queen. The impact of this selection pressure on the next generation and measured performance traits could then be documented. Some breeding programs that target varroa resistance mechanisms in honeybees use queen insemination with a single selected male to reduce genetic variation on the paternal side to a minimum, thus ensuring strong selection for key traits [50,51]. This approach has important limitations, however, since the quantity of sperm in the queen’s spermatheca is minimal (approximately 1 µL per drone), which reduces fecundity and impacts the pheromones she releases. This in turn has a strong effect on interactions among the workers, colony cohesion and performance [52,53].

During this study, winter losses were higher than summer losses for all colonies (with both inseminated and naturally fertilized queens). However, we measured an additional 6.3% total colony loss with inseminated queens. Furthermore, 73% of the summer losses in colonies with inseminated queens were linked to a queen problem. These losses, considered abnormal, could be linked to a failure of our insemination technique [22]. Indeed, many factors strongly impact the success of insemination, including total asepsis, mucus presence in the syringe and time needed to complete the insemination [23,54,55]. The losses in our study could also be linked to a genetic weakness within the colony caused by the insemination of queens by related drones [56,57,58,59]. Low genetic diversity within a colony reduces the behavioral diversity of the workers and limits population growth, with repercussions on the use of environmental resources and even on resistance to diseases, parasites and environmental fluctuations [3,56,60,61,62,63,64,65,66,67].

## 5. Conclusions

This study shows that the use of instrumental insemination is not effective in increasing genetic gains for several performance traits in only one generation of honeybees. However, this investigation allowed us to examine new avenues for maintaining the precision in breeding values that accompanies total control of reproduction via instrumental insemination, while accelerating genetic gains in performance within our selection program. Our findings show that instrumental insemination can be a useful and efficient tool for obtaining total reproductive control within a genetic selection program, with the caveat that it must be used repeatedly over several generations.

## Figures and Tables

**Figure 1 insects-14-00301-f001:**
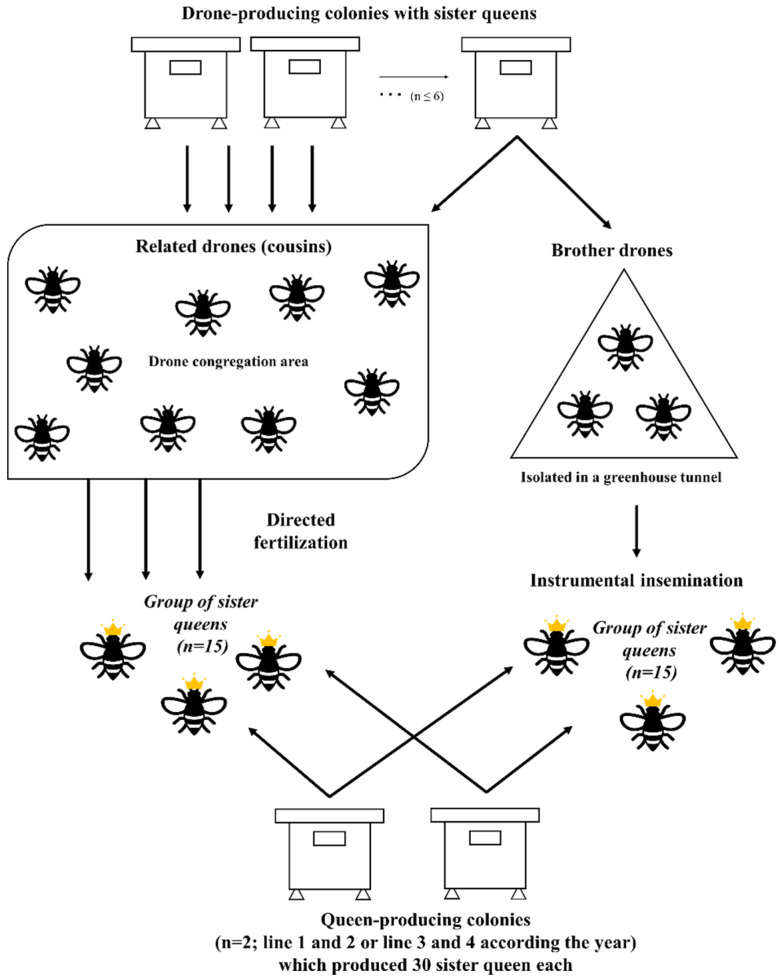
Experimental design for each year of the project.

**Figure 2 insects-14-00301-f002:**
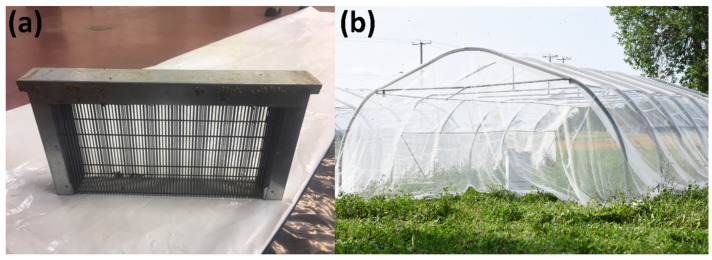
(**a**) Queen excluder cage (photo: Andrée Rousseau); (**b**) small drone hive placed in a greenhouse tunnel covered with a market garden shade net (photo: Georges Martin).

**Figure 3 insects-14-00301-f003:**
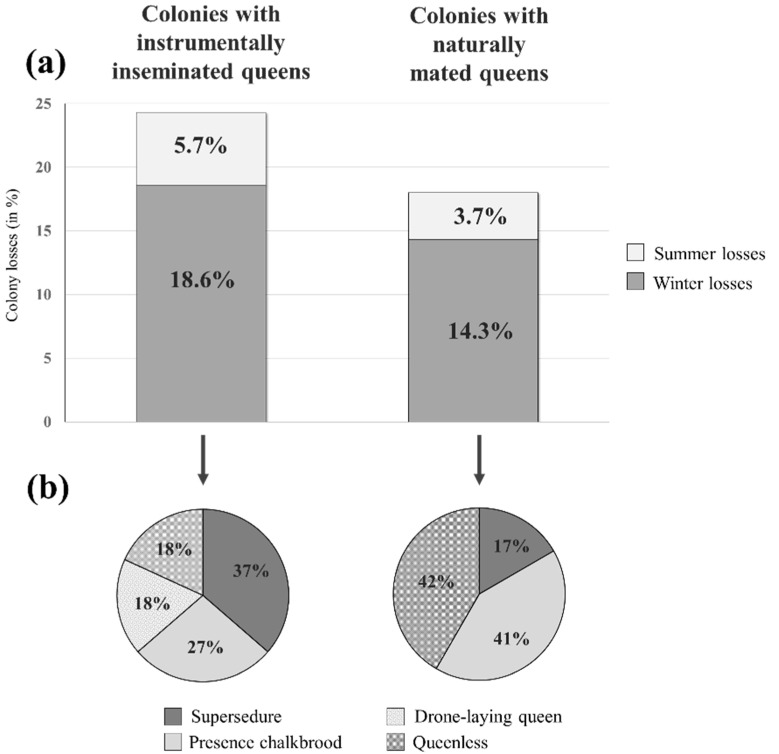
(**a**) Summer and winter percentages of colony losses (including 2019 and 2020); (**b**) proportions of identified causes of total summer colony losses for 2019 and 2020.

**Figure 4 insects-14-00301-f004:**
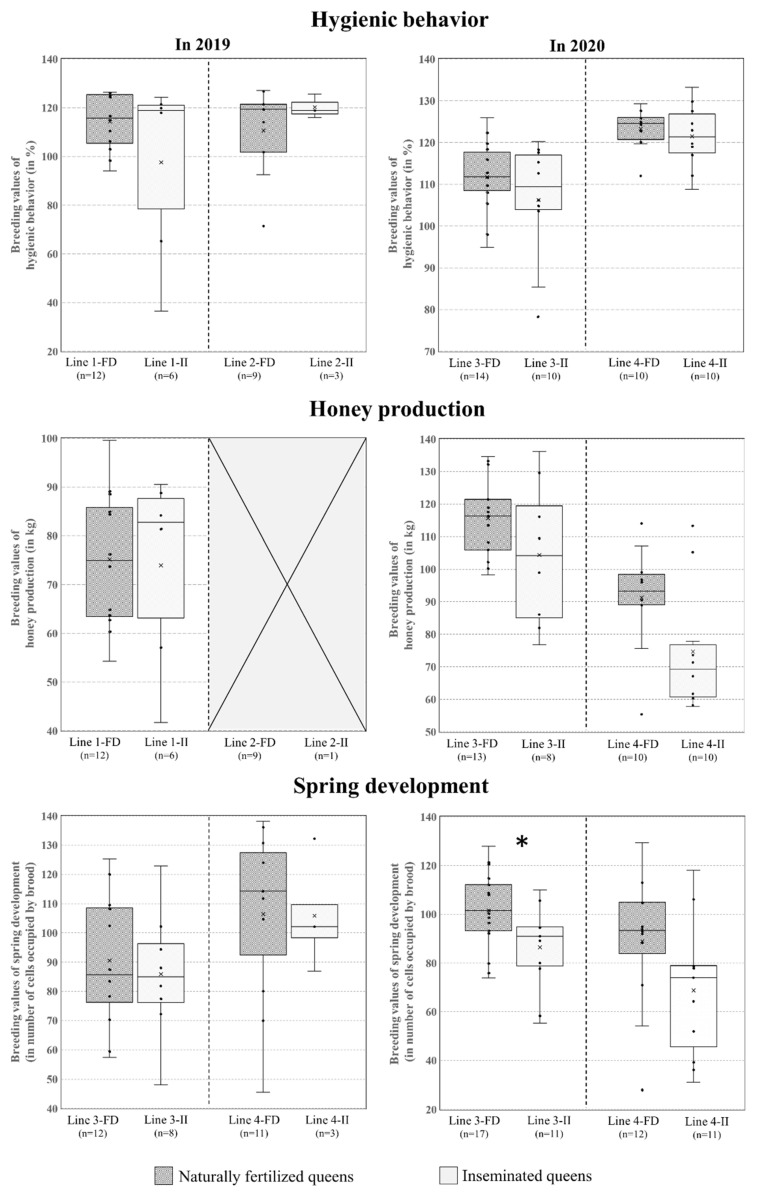
Comparison of breeding values for three traits: hygienic behavior, honey production and spring development, between colonies with inseminated queens and those with naturally fertilized queens (level of significant difference between the two groups of queens: * *p* < 0.05).

**Table 1 insects-14-00301-t001:** Selection index and breeding values of selected mother colonies used to produce young queens in 2018 and 2019. The selection index combines the breeding values of three selection criteria: hygienic behavior, honey production and spring development, at percentages of 50, 30 and 20, respectively.

Queen-Producing Colonies		Selection Index	Breeding Value of Hygienic Behavior	Breeding Value of Honey Production	Breeding Value of Spring Development
2018	Mother colony for line 1	178.5	185.9	116.3	133.9
	Mother colony for line 2	130.2	156.5	87.8	81.9
2019	Mother colony for line 3	154.7	132.8	156.3	144.7
	Mother colony for line 4	153.4	132.8	163.0	131.5

**Table 2 insects-14-00301-t002:** Selection index and breeding values of selected sister colonies used to produce many drones for our mating station in 2018 and 2019 (average ± SE). Two selection indexes are presented: a selection index of the mother colony of these sister colonies and an average selection index of these sister colonies. Each selection index combines the breeding values of three selection criteria: hygienic behavior, honey production and spring development, at percentages of 50, 30 and 20, respectively.

Drone-Producing Colonies for Natural Mating		Selection Index of Mother Colony of Sister Colonies Producing Drones	Average Selection Index of Sister Colonies Producing Drones	Breeding Value of Hygienic Behavior	Breeding Value of Honey Production	Breeding Value of Spring Development
2018	6 sister colonies	104.3	93.0 ± 5.0	93.8 ± 5.0	97.2 ± 4.6	95.3 ± 11.1
2019	8 sister colonies	148.2	119.3 ± 4.0	123.4 ± 3.9	108.5 ± 5.2	100.1 ± 4.9

**Table 3 insects-14-00301-t003:** Selection index and breeding values of selected mother colonies used to produce drones for sperm collection / instrumental insemination in 2018 and 2019. Two selection indexes are presented: a selection index of the mother colony of colonies used and a selection index of colonies used. The selection index combines the breeding values of three selection criteria: hygienic behavior, honey production and spring development, at percentages of 50, 30 and 20, respectively.

Drone-Producing Colonies for Instrumental Insemination		Selection Index of Mother Colony of Drone-Producing Colony	Selection Index of Drone-Producing Colony	Breeding Value of Hygienic Behavior	Breeding Value of Honey Production	Breeding Value of Spring Development
2018	1 drone-producing colony	104.3	106.4	116.6	94.8	88.6
2019	1 drone-producing colony	148.2	136.0	127.8	138.1	125.6

## Data Availability

Data set is available upon request to the corresponding author.

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
