# Peer review of "Observation of Genetic Gain with Instrumental Insemination of Honeybee Queens"

_insects, 2023, doi:10.3390/insects14030301_

Round 1

Reviewer 1 Report

The paper entitled “Observation of genetic gain with instrumental insemination of honeybee queens”, by Maucourt et al. reports a study to test the efficacy of instrumental insemination (as compatred to natural mating) of honeybee queens using the BLUP-animal selection model in a breeding program, aiming to accelerate genetic gain at the F1 generation for three traits of interest to the Canadian beekeeping industry: hygienic behavior, honey production and spring development.

Although the results do not demonstrate to be effective in increasing genetic gains for the three traict considered, they indicate that instrumental insemination can be a useful tool to achieve  total reproductive control within a genetic selection program.

The article is well written. Particularly, the "materials and methods" are accurately described and in the discussion, data are well argumented and properly accompained by literature citations. Thus, I suggest the paper merits to be published in “Insects”.

Author Response

Dear Reviewer,

Thank you for taking the time to review our manuscript. We are very grateful for the comments and corrections that you mentioned. These pertinent remarks have allowed us to improve the quality of the manuscript.

Sincerely,

Ségolène Maucourt

Reviewer 2 Report

This paper is well-written, scientifically sound, and worthy of publication. Its use of genetic protocols improve upon previous reports in the literature. My only caveat is that the major conclusions are based upon the efficacy of the artificial insemination technique. Improvements in that technique to match the success of insemination under natural conditions might change the conclusions. However, the authors did note that even though artificial insemination ascurrently practiced may have problems, it is still a way of more tightly controlling the genetic outcomes of mating. Suggestions for improvement are noted on the manuscript.

Author Response

Dear Reviewer,

Thank you for taking the time to review our manuscript. We are very grateful for the comments and corrections that you mentioned. These pertinent remarks have allowed us to improve the quality of the manuscript.

Below are the responses to all of your specific comments. 

Sincerely, 

Segolene Maucourt

Specific comments: 

Point 1 : l.21 Insert « artificially »

Response 1: We agree, “artificially” has been added to the line 21.

Point 2: “they” not their

Response 2: We agree, “their” has been replaced by "they" to the line 24.

Point 3: “mating site”?

Response 3: We agree, “fertilization site” has been replaced by "mating site" to the line 69.

Point 4: A justification for these percentages would be helpful

Response 4: Information has been added to the lines 113-115.

Point 5: Clarify: “twice”:  I assume once in 2018 and once in 2019, NOT twice in each year?

Response 5: Sentence has been modified from line 119 to 120.

Point 6: I don’t find the lower part of this diagram to be transparent. I suggest showing 2 queen-producing hives, and then showing arrows of 15 going to each insemination treatment from each of the 2 hives.

Response 6: The requested modifications have been made on the Figure 1.

Point 7: I suggest: Experimental design for each year of the project.

Response 7: The title of the Figure 1 has been modified to the line 125.

Point 8: These lines are undefined. Maybe if you added the numbers 1 (3) and 2 (4) to the above diagram for the 2 colonies that I suggested, that would identify them.

Response 8: Information has been added to the Figure 1.

Point 9: This is Table 2 (correct in text)

Response 9: It is an error of inattention. We have modified this error in the title to the line 156 and the text has been checked.

Point 10: Table 3

Response 10: It is an error of inattention. We have modified this error in the title to the line 169 and the text has been checked.

Point 11: I wonder if semen deteriorates during overnight storage. I work with moth sperm, and they partially deteriorate over such long storage in vitro. I don’t have access to the references given, and the DOI for #23 gives an error message, so don’t know if viability of overnight storage has been assessed for bee semen.

Response 11: We agree, the DOI link for citation 23 has been changed to the line 422. Drone seed is viable for several months at room temperature in appropriate capillaries (Collins, A. M. 2000. Survival of honeybee (Hymenoptera: Apidae) spermatozoa stored at above-freezing temperatures. J. Econ. Entomol. 93: 568–571.). This is what was done in this protocol, the seed was harvested the day before the inseminations, so we don’t think that this played a role in the results presented in this article.

Reviewer 3 Report

Great paper!  Covered all the bases.  Important findings for our industry.

The only thing that I would have added is that overall colony performance is the result of the combined traits of a "team" of genetically-distinct patrilines of workers, which may individually exhibit critical behaviors.  A perfect example of Aristotle's “The whole is greater than the sum of its parts.” 

Author Response

Dear Reviewer,

Thank you for taking the time to review our manuscript. We are very grateful for the comments and corrections that you mentioned. These pertinent remarks have allowed us to improve the quality of the manuscript.

Sincerely,

Segolene Maucourt